# Automatic Annotation Diagnostic Framework for Nasopharyngeal Carcinoma via Pathology–Fidelity GAN and Prior-Driven Classification

**DOI:** 10.3390/bioengineering11070739

**Published:** 2024-07-22

**Authors:** Siqi Zeng, Xinwei Li, Yiqing Liu, Qiang Huang, Yonghong He

**Affiliations:** 1Medical Optical Technology R&D Center, Research Institute of Tsinghua, Pearl River Delta, Guangzhou 510700, China; zengsq@tsinghua-gd.org; 2School of Computer, Electronics and Information, Guangxi University, Nanning 530004, China; 2012270246@st.gxu.edu.cn; 3Institute of Biopharmaceutical and Health Engineering, Shenzhen International Graduate School, Tsinghua University, Shenzhen 518055, China; liuyiqin20@mails.tsinghua.edu.cn; 4Shenzhen Shengqiang Technology Co., Ltd., Shenzhen 518055, China; hq@sqray.com

**Keywords:** nasopharyngeal carcinoma, deep learning, pathological diagnosis framework, digital staining, pathology–fidelity

## Abstract

Non-keratinizing carcinoma is the most common subtype of nasopharyngeal carcinoma (NPC). Its poorly differentiated tumor cells and complex microenvironment present challenges to pathological diagnosis. AI-based pathological models have demonstrated potential in diagnosing NPC, but the reliance on costly manual annotation hinders development. To address the challenges, this paper proposes a deep learning-based framework for diagnosing NPC without manual annotation. The framework includes a novel unpaired generative network and a prior-driven image classification system. With pathology–fidelity constraints, the generative network achieves accurate digital staining from H&E to EBER images. The classification system leverages staining specificity and pathological prior knowledge to annotate training data automatically and to classify images for NPC diagnosis. This work used 232 cases for study. The experimental results show that the classification system reached a 99.59% accuracy in classifying EBER images, which closely matched the diagnostic results of pathologists. Utilizing PF-GAN as the backbone of the framework, the system attained a specificity of 0.8826 in generating EBER images, markedly outperforming that of other GANs (0.6137, 0.5815). Furthermore, the F1-Score of the framework for patch level diagnosis was 0.9143, exceeding those of fully supervised models (0.9103, 0.8777). To further validate its clinical efficacy, the framework was compared with experienced pathologists at the WSI level, showing comparable NPC diagnosis performance. This low-cost and precise diagnostic framework optimizes the early pathological diagnosis method for NPC and provides an innovative strategic direction for AI-based cancer diagnosis.

## 1. Introduction

Nasopharyngeal carcinoma (NPC) is a malignant epithelial tumor primarily occurring in the pharyngeal recess of the nasopharynx, yet it exhibits a notably high incidence in certain Asian regions. The early stages of NPC often present non-specific symptoms, such as nasal congestion, tinnitus, and hearing loss, which make early detection challenging. Consequently, many patients are diagnosed at the advanced stage of the disease [1,2]. Therefore, early diagnosis is crucial for improving the survival and prognosis of NPC patients, as the five-year survival rate for those diagnosed early can exceed 90%, significantly higher than that of patients diagnosed at the advanced stage [3,4,5]. Currently, the gold standard for the clinical detection of NPC involves an endoscopic examination combined with a pathological analysis of biopsies [6,7,8].

The World Health Organization (WHO) classifies NPC into three pathological subtypes: keratinizing squamous cell carcinoma, non-keratinizing carcinoma, and basaloid squamous cell carcinoma [9]. In endemic regions, non-keratinizing nasopharyngeal carcinoma (NK-NPC) is the most common subtype, with a 100% Epstein–Barr virus (EBV) infection rate among patients [10,11]. Therefore, EBV testing is crucial in the diagnosis of NPC. By utilizing EBV-encoded small RNA (EBER) for in situ hybridization (ISH), EBER–ISH serves as a critical method for detecting EBV infection in tumor cells [12,13,14,15,16,17,18], and it plays a central role in the pathological diagnosis of NPC. As shown in the EBER examples in Figure 1, only the nuclei of tumor cells are stained positive for EBER (appearing as blue or blue-brown), while all non-NPC cells are stained negative (appearing as pink).

With the advancement of computer-assisted diagnostic technologies, the application of deep learning in pathological image analysis has become increasingly prevalent. Recently, several researchers have leveraged deep learning in conjunction with pathological images to diagnose NPC. For instance, Chuang et al. employed deep convolutional neural networks for the first time to automatically diagnose NPC based on 726 biopsy samples, effectively distinguishing between cancerous and non-cancerous cases [19]. Subsequently, Diao et al. utilized the Inception-v3 model to diagnose chronic nasopharyngeal inflammation, lymphatic hyperplasia (LHP), and NPC, outperforming junior and mid-level pathologists in test sets [20]. Building on this work, they further developed a transformer-based network model using 227 NPC cases collected between 2004 and 2018, which successfully enabled precise segmentation of NPC tumor regions [21]. Additionally, W. S. H. M. W. Ahmad et al. utilized the DenseNet architecture to analyze seven whole slide images (WSIs) from two hospitals, successfully automating the classification of nasopharyngeal inflammation, LHP, NPC, and normal tissues [22]. Lin et al. have developed a low-cost method for diagnosing NPC by utilizing EBER-stained images to assist in annotating H&E images, achieving a diagnosis process that does not require direct participation from pathologists [23].

A review of previous studies on AI-based NPC diagnosis indicates that model training typically necessitates the manual annotation of H&E images by pathologists. However, NPC’s complex tumor microenvironment [24,25,26,27,28], exemplified by lymphocyte infiltration (as illustrated in Figure 1), complicates the pathological diagnosis. Consequently, constructing a high-quality H&E-annotated training dataset incurs significant costs. Although some scholars have proposed using other staining methods to assist H&E annotation to improve efficiency, subjective annotation by non-pathologists may reduce data quality and increase semantic ambiguity in model training, thereby affecting the diagnostic effect of the model. Given the critical role of EBER staining specificity in diagnosis, it is feasible to leverage the distinctive image features of NPC tumor cells compared to other cells, such as lymphocytes, in H&E images for facilitating a digital staining conversion to EBER images. This approach could potentially allow for the indirect diagnosis of H&E images based on the specificity of EBER staining, thus potentially avoiding complex and subjective H&E image annotation while enhancing diagnostic accuracy.

Virtual staining technology utilizes deep learning to achieve stain-to-stain transformation. Unlike traditional staining methods, virtual staining eliminates the need for physical slide preparation and repeated staining processes, enabling the conversion between different staining styles. This approach not only enhances diagnostic efficiency and significantly reduces costs but also allows pathologists to switch between various staining perspectives quickly, thus accelerating the acquisition of diagnostic information. GANs commonly serve as the primary technology in virtual staining applications. Bai et al. employed a conditional generative adversarial network to transform autofluorescence microscopy images of breast tissue into HER2 stained images [29]. Li et al. introduced a saliency constraint into the CycleGAN model, successfully converting immunofluorescence images of colon tissue into H&E-stained images using unpaired training data [30]. De Haan et al. utilized paired images from kidney biopsies to convert H&E-stained images into special stained images successfully [31]. Additionally, Liu and colleagues enhanced CycleGAN with pathology properties and consistency constraints, achieving a transformation from H&E-stained breast and neuroendocrine tissues to the Ki-67 staining style, ensuring the consistency of pathological properties across different domains [32].

To overcome the limitations of current AI-based NPC diagnostic models, this study introduces a novel diagnostic framework named PFPD. This framework contains a generative adversarial network model (PF-GAN) incorporating pathology–fidelity constraints and a prior-driven image classification system (PD-CS). PF-GAN effectively transforms H&E images into EBER images to complete digital staining. Subsequently, PD-CS leverages the specificity of EBER staining and pathological prior knowledge to accurately classify these images, achieving NPC diagnosis. The high accuracy of PD-CS in classifying EBER images enables the automatic annotation of PF-GAN’s training data, significantly reducing annotation costs and minimizing subjective biases. The primary contributions of this work include:This work introduces a novel AI-based framework for diagnosing NPC, which initially achieves accurate digital staining of H&E images to EBER images. Subsequently, the framework utilizes pathological prior knowledge to diagnose NPC effectively.This work designs a generative adversarial network to achieve feature alignment from H&E to EBER images by integrating pathology–fidelity constraints. Furthermore, the model establishes standardized staining for EBER images, effectively addressing the staining overflow commonly encountered in traditional EBER staining techniques.This paper develops a prior-driven image classification system (PD-CS) that leverages pathological prior features to establish a set of threshold criteria. These criteria are designed to accurately classify EBER images based on specific pathological indicators.During the model training process, this paper facilitates the automatic labeling of the training dataset, significantly enhancing the development efficiency of AI-based NPC models.

## 2. Methods

### 2.1. Overview of NPC Diagnosis Using the PFPD Framework

Figure 2 illustrates an overview of PFPD in NPC diagnosis. The cooperation between digital taining and PD-CS is the core of diagnosis. Initially, the PFPD utilizes a GAN network to transform H&E patches into EBER patches, completing digital staining. Subsequently, PD-CS leverages the specificity of EBER staining and pathological prior knowledge to classify the digital-stained EBER patches. This classification process indirectly completes the classification of H&E patches. Ultimately, the outputs of PFPD include the diagnostic results for NPC, digital-stained EBER WSI, and the corresponding mask of the tumor regions.

Digital staining utilizes image translation by learning the feature mapping relationship between H&E images and EBER images through a GAN. This mapping ensures that NPC tumor cells appear EBER-positive after digital staining, while non-NPC tumor cells appear EBER-negative. In the digital staining process, H&E patches are extracted from the foreground area of the H&E WSI using a sliding window technique. These patches then serve as inputs to the GAN model, which uses its learned mapping to achieve digital staining from H&E to EBER. The digital-stained EBER patches are reconstructed into the digital-stained EBER WSI and serve as inputs for the PD-CS. To enhance the GAN model’s capability in learning the mapping relationship between H&E and EBER images, we propose an improved model named PF-GAN, with specific details described in Section 2.2.

The principal objective of PD-CS is to exploit staining specificity and pathological prior knowledge for a threshold-based classification of digital-stained EBER images, which indirectly classifies the original H&E patches. Initially, the pixel-level decision classifies pixel values in digital-stained EBER patches to generate binary masks, identifying EBER-positive regions. Then, the patch-level decision evaluates each mask’s largest connected domain based on the tumor cell pixel size threshold. EBER patches corresponding to masks that meet the threshold are labeled as positive (1), while EBER patches corresponding to masks that do not meet the threshold are labeled as negative (0). The WSI-level decision assesses the count of positive labels against the positive image count threshold to diagnose the entire original HE WSI as either NPC or non-NPC. Additionally, these masks are reconstructed into masks of the tumor regions corresponding to the H&E WSI. For the detailed technical specifics and threshold choices based on pathological knowledge, refer to Section 2.3. Ultimately, PFPD completes the diagnosis of NPC from the H&E WSI.

### 2.2. Pathology–Fidelity Generative Adversarial Network

Despite the tremendous success of GANs in style transfer for natural images [33,34,35,36], their application to pathological images still has limitations. In digital staining with GANs, generated images pass discriminator verification but fail to preserve the original pathological features, compromising pathological consistency. However, pathological consistency and image generation quality are equally important in digital staining. In the digital staining process from H&E to EBER, pathological consistency refers to ensuring that NPC tumor cells in H&E images correctly exhibit EBER-positive image features (such as color and shape) when converted to EBER images, while non-NPC tumor cells maintain EBER-negative image features. To achieve pathological consistency, we introduced pathology–fidelity constraints into the Contrastive Unpaired Translation (CUT) [37] model named PF-GAN. Figure 3 illustrates the structure of PF-GAN.

The primary goal of digital staining is to use models to learn image feature mappings under different staining conditions, achieving image conversion from the source domain $X\subseteq R^{H\times W\times C}$ to the target domain $Y\subseteq R^{H\times W\times 3}$. The PF-GAN model, inspired by the CUT model, employs a generator and a discriminator and integrates adversarial generation with contrastive learning to train on unpaired instances; X={x∈X} and Y={y∈Y}. Distinct from the CUT model, PF-GAN incorporates a classifier during training and introduces pathology–fidelity constraints, which guide the generator in maintaining pathological consistency throughout the image generation process. The loss function for the PF-GAN, denoted as L, is defined as follows: (1)$$\begin{array}{cc} L= & \;L_{GAN}\left(G,D,X,Y\right)+L_{PatchNCE}\left(G,H,X\right)\\ \; & \;+L_{PatchNCE}\left(G,H,Y\right)+\lambda_{PF}L_{PF}\left(G,cls,Y\right) \end{array}$$
where $L_{GAN}$ represents the adversarial loss, employed to train the generator *G* to produce visually convincing images, while training the discriminator *D* to differentiate between real and generated images. Minimizing this loss enables the generator to deceive the discriminator, effectively generating more realistic images. The PatchNCE loss utilizes a contrastive learning mechanism to extract and compare features from the local areas of source images and generated images. This approach enhances the balance between content fidelity and style adaptability, enabling high-quality image translation without paired training data. $L_{PatchNCE}\left(G,H,X\right)$ uses a two-layer MLP network [38] as a feature extractor *H* to extract features from local regions of the source image *X* as positive samples and compare them with corresponding regions in the generated image G(X). It also randomly selects other regions in the source image as negative samples. Maximizing the similarity of positive samples and minimizing the similarity of negative samples ensures that the generated image retains the local structure and content consistency with the original image. Similarly, LPatchNCE(G,H,Y) ensures that the features of the target domain image *Y* are correctly preserved and presented in the image generation by performing contrastive learning in *Y* and G(Y). In PF-GAN, the original settings of the CUT model are retained, where the weights of $L_{PatchNCE}\left(G,H,X\right)$ and $L_{PatchNCE}\left(G,H,Y\right)$ are both set to 1.

$L_{PF}\left(G,cls,Y\right)$ represents the pathology fidelity constraint. During training, a single-layer MLP network, denoted as cls, is used as a classifier to train the encoder $G_{enc}$ within the generator to classify the target domain. This process enables the generator to learn global pathological features, ensuring the generated images maintain pathological consistency. The specific definition of the fidelity constraint is as follows: (2)$$L_{PF}\left(G,cls,Y\right)=E_{y\in Y}\left(∥cls\left(G_{enc}\left(y\right)\right)-l_{y}{∥}_{1}\right)$$
where $l_{y}$ denotes the real label of *y*, and $cls(G_{enc}\left(y\right))$ represents the predicted label by the encoder. The core purpose of the PatchNCE loss function is to ensure local consistency through contrastive learning, thereby enhancing the fidelity of local details (such as texture) in the generated images. In contrast, pathology–fidelity loss emphasizes the global pathological features of the generated images, including crucial characteristics like the color and shape of tumor cells. The pathology–fidelity loss extracts global pathological features from the target domain, which are shared during the image generation process by training the encoder’s classification capability. It indirectly influences the generation process, ensuring pathological consistency in the generated images. The combination of the classification task and contrastive learning provides dual supervisory signals that optimize the encoder’s weights in the generator, enabling the model to maintain consistency in both the local details and the global pathological features during the generation process and enhancing pathological fidelity in digital staining tasks. Given that image generation quality and pathological consistency are equally crucial in digital staining, i.e., both local detail features and global pathological features are equally significant, $\lambda_{PF}$ is set to 1. Finally, the objective function L of PF-GAN is: (3)$$\begin{array}{c} L=L_{CUT}+L_{PF} \end{array}$$

Subsequent experimental results demonstrate that with the pathology–fidelity constraint, PF-GAN generates high-quality images with pathological consistency.

### 2.3. The Prior-Driven Image Classification System

Although the PF-GAN can already classify EBER images using the integrated classifier, we have further developed a PD-CS to enhance the flexibility and scalability of the entire NPC pathological diagnostic framework. The system utilizes the specificity of EBER staining and pathological prior knowledge, combined with image processing technology. It enables the accurate classification of EBER images and ultimately allows for diagnosing the H&E WSI input of the framework.

Figure 4 illustrates the diagnostic process of the PD-CS, which consists of three main modules: pixel-level decision, patch-level decision, and WSI-level decision. Through the model described above, from selecting thresholds based on EBER specificity to choosing pathology connectivity thresholds and up to the comprehensive diagnostic aggregation for WSI, we have realized an end-to-end NPC diagnosis from pixels to WSI. In order to explore the threshold selection for the relevant modules of the system and improve its robustness, we randomly sample 1000 positive and 1000 negative EBER images from the training dataset. Using these 2000 images, we conduct a study aimed at enhancing the reliability of each submodule within the system.

In the pixel-level decision module, we analyzed EBER image channels and noted higher pixel values in the R channel for negative images, as shown in Figure 5. The histograms in Figure 6 further demonstrated that negative images cluster in high-intensity ranges (greater than 200), whereas positive ones occupy low-intensity areas (less than 100), showcasing the R channel’s discriminative efficiency. Utilizing the specificity of EBER staining, the pixel-level decision module achieves pixel-level thresholding. This module first filters out background pixels and then extracts the R channel pixel values of the foreground. Pixels with values less than 100 are classified as positive, and those with higher values are negative.

In the patch-level decision module, based on the pixel-level classification results, corresponding masks are constructed, where ‘1’ indicates positive, and ‘0’ denotes negative and background pixels. Although patch-level masks are obtained, the presence of positive pixels alone does not directly determine the image’s positivity due to noise interference. In this study, we analyze 1000 selected EBER-positive images. Assisted by experienced pathologists, we identify and measure the pixel size of the smallest tumor cell in each image. By counting the pixels within the bounding rectangles of these cells, we establish that the average pixel count is 125. This result facilitates an approximate assessment of tumor cell dimensions. Based on the pathological prior knowledge of tumor cell pixel size, we utilize the size of connected regions within these masks to judge the positivity of the images. To reduce estimation error and improve sensitivity, we set the threshold at 100 pixels. If the largest connected region of positive pixels within a mask exceeds this threshold, the respective EBER patch is classified as positive; otherwise, it is classified as negative, thus completing the patch-level classification process.

In the WSI-level decision module, the pathologist’s diagnostic criteria determine that the case is classified as nasopharyngeal carcinoma (NPC) if tumor cells are observed in the EBER images. Based on the pathological prior knowledge regarding tumor cell counts, we gather the number of positive EBER images to establish the final WSI-level classification criteria. If a positive EBER image is identified, the entire WSI is diagnosed as NPC; otherwise, it is categorized as non-NPC. This criterion follows the pathologist’s usual decision-making process.

## 3. Experiments

### 3.1. Datasets

The data for this study were sourced from Peking University Shenzhen Hospital, encompassing all NPC pathology slides collected between 2020 and 2023. The dataset comprises WSIs from 232 cases, including two types of staining (H&E and EBER). All slides were digitized using the SQS600P-0073 high-throughput scanner of Shenzhen Shengqiang Technology Co., Ltd. (Shenzhen, China). The H&E and EBER-stained WSIs were subdivided into overlapping image patches of 256 × 256 pixels at a resolution of 0.1036 μm/pixel. This high resolution significantly enhanced the capture of detailed cellular and tissue structures in the images, thereby optimizing data quality.

To ensure the effectiveness and scientific rigor of the model training [39], this study selected 44 representative cases (28 positive, 16 negative) for training, with H&E-stained WSIs segmented into 71,216 patches (40,525 positive, 30,691 negative) and EBER-stained WSIs into 83,928 patches (41,482 positive, 42,446 negative). The remaining 188 cases (124 positive and 64 negative) were used for testing, with the H&E-stained WSIs segmented into 77,159 patches to evaluate the model’s performance in practical applications.

This paper trained all models using the same dataset to ensure fairness. Specifically, both the source domain (H&E images) and the target domain (EBER images) used the same dataset to train all GANs. In the comparative experiments between PFPD and the fully supervised models, the fully supervised models were based on the H&E dataset from the GAN training set. These H&E images, annotated by pathologists, included a total of 40,525 NPC and 30,691 non-NPC images. There is no obvious skew in the training data. Training all models on the same H&E dataset ensured fairness in comparison and evaluation. Regarding the data quality, the 44 cases used for training were carefully selected by pathologists, ensuring high representativeness and comprehensiveness. Compared to random case selection, this careful selection process enhances the robustness of model training.

### 3.2. Automatic Annotation in Training PF-GAN

Due to the introduction of pathology–fidelity constraints, PF-GAN requires labeled EBER images to train the generator to extract global pathological features. Considering the high consistency between PD-CS and pathologists in classifying EBER images, we can leverage the pixel-level and patch-level modules of PD-CS to automatically and efficiently annotate EBER images for training the PF-GAN model. The specific process is illustrated in Figure 7. The EBER training images are input into the PD-CS system, where the pixel-level module uses the pixel value threshold to distinguish positive and negative pixels, and the patch-level module applies the tumor cell pixel size threshold to assign class labels to the EBER training images (1 for positive, 0 for negative). This automated process provides labeled EBER training images for PF-GAN training. It eliminates the need for manual annotation by pathologists, reducing the time-consuming aspect associated with manual labeling and improving the efficiency of NPC diagnostic model development.

It is worth mentioning that another advantage of PF-GAN model training is that it does not require paired data. The model achieves the image conversion of tumor cells and non-NPC cells from the H&E staining domain to the EBER staining domain by learning the mapping relationship between unpaired H&E and EBER images. By combining the automatic annotation technology of PD-CS with the PF-GAN model’s ability to process unpaired data, PFPD addresses the bottleneck of manual annotation by pathologists in the development of previous AI-based NPC diagnostic models, significantly reducing the workload of pathologists during the model development process. Furthermore, compared to the uncertain time cost of manual annotation, PFPD’s automatic annotation is more stable. Due to the rapid development of existing hardware resources, automatic computer-assisted annotation is significantly faster than manual annotation.

### 3.3. Experimental Settings

The experiments were conducted on an NVIDIA GeForce RTX 4090 GPU (manufactured by NVIDIA Corporation, located in Santa Clara, CA, USA) with 24 GB of memory. The system was equipped with 251 GiB of RAM. GANs utilized the Adam optimizer for model optimization and conducted 200 epochs with a batch size of 4. Among them, PF-GAN employed the Resnet-based generator, which includes 9 Resnet blocks interspersed with several downsampling and upsampling operations. The initial learning rate was set at 0.0002, with the momentum parameters ($\beta_{1}$ and $\beta_{2}$) set at 0.5 and 0.999, respectively. The learning rate began to decay from the midpoint of the training duration.

### 3.4. Evaluation Metrics

In this study, the performance of different GANs was evaluated from two perspectives: generation quality evaluation and accuracy analysis.

The image quality evaluation was based on RMSE [40], PSNR [41], and SSIM [42,43] metrics. Mathematically, these metrics are defined as follows: (4)$$\mathrm{RMSE}=\sqrt{\frac{1}{mn}\sum_{i=1}^{m}\sum_{j=1}^{n}{\left(I\left(i,j\right)-K\left(i,j\right)\right)}^{2}}$$
(5)$$\mathrm{SSIM}\left(I,\hat{I}\right)=\frac{\left(2\mu_{I}\mu_{\hat{I}}+C_{1}\right)\left(2\sigma_{I\hat{I}}+C_{2}\right)}{\left(\mu_{I}^{2}+\mu_{\hat{I}}^{2}+C_{1}\right)\left(\sigma_{I}^{2}+\sigma_{\hat{I}}^{2}+C_{2}\right)}$$
(6)$$\mathrm{PSNR}=20\log_{10}\left(\frac{{\mathrm{MAX}}_{I}}{\mathrm{RMSE}}\right)$$

The accuracy analysis was conducted using accuracy [44], precision [45], recall [46], specificity [47] and F1-Score [48]. Mathematically, these metrics are defined as follows: (7)$$\mathrm{Accuracy}=\frac{TP+TN}{TP+TN+FP+FN}$$
(8)$$\mathrm{Precision}=\frac{TP}{TP+FP}$$
(9)$$\mathrm{Recall}=\frac{TP}{TP+FN}$$
(10)$$\mathrm{Specificity}=\frac{TN}{TN+FP}$$
(11)$$F1\text{-}\mathrm{Score}=\frac{2\times \mathrm{Precision}\times \mathrm{Recall}}{\mathrm{Precision}+\mathrm{Recall}}$$

## 4. Results

### 4.1. Pathological Fidelity Evaluation via Qualitative Analysis

For pathological fidelity evaluation, the PFPD framework uses different GANs (CycleGAN, CUT, PF-GAN) as the digital staining backbone for comprehensive qualitative analysis. CycleGAN [49] enables transformation between two image domains without paired samples by introducing a cycle consistency loss. CUT employs contrastive learning principles to achieve style transfer in unpaired images by maximizing mutual information between image patches. PF-GAN is the novel model proposed in this study. Subsequently, digital EBER-stained images were compared with real H&E and EBER-stained images at the WSI level and within organizational regions.

In Example 1 of Figure 8, for negative WSIs, CycleGAN and CUT exhibit a certain degree of over-staining. This over-staining can be attributed to the absence of pathology–fidelity loss in these GANs, resulting in converted images that lack authenticity and fail to preserve the pathological consistency of the original H&E-stained images. In contrast, PF-GAN successfully converts to the correct EBER-negative staining, demonstrating that pathology–fidelity loss ensures consistent stain conversion, with non-NPC cells in the H&E-stained images remaining negative in the converted EBER images.

In Example 2 of Figure 8, for positive WSIs, CycleGAN, CUT, and PF-GAN correctly convert tumor cells in the H&E images to positive staining in the EBER images. Additionally, PF-GAN preserves the negative staining of non-NPC cells distributed within the tumor regions. This ability to retain non-NPC cell characteristics is achieved by incorporating pathology–fidelity loss during model training. The constraints imposed by this loss ensure a standard conversion process, maintaining the consistency and authenticity of pathological features. The superior performance of PF-GAN in pathological fidelity significantly enhances its applicability in practical use.

Figure 9 illustrates the staining performance of GANs within the organizational regions. Although CycleGAN and CUT successfully convert tumor cells from H&E staining to positive EBER staining (Example 3 and Example 4), their performance in non-NPC cell regions is suboptimal. They exhibit pronounced over-staining and a tendency towards simplistic color style conversions, undermining their pathological fidelity (Example 1 and Example 2). In contrast, PF-GAN accurately converts tumor cells to positive staining (Example 3 and Example 4) and correctly converts non-NPC cells to negative areas in EBER staining (Example 1 and Example 2). This capability significantly enhances PF-GAN’s performance in pathological fidelity, making it more advantageous for practical applications.

In summary, the qualitative analysis of pathological fidelity underscores the effectiveness of incorporating pathology–fidelity loss to maintain consistency and authenticity in staining. PF-GAN demonstrates the best staining performance in qualitative analysis at the WSI level and within organizational regions.

### 4.2. Evaluation of PF-GAN Digital Staining in Difficult Cases

In the pathological diagnosis of NPC, due to the complexity of the tumor microenvironment, especially the infiltration of lymphocytes, it is often challenging to rely solely on H&E images for diagnosis, which may lead to the extended diagnosis process and reduced accuracy. To address this, this study selected two representative difficult cases with lymphocytic infiltration to evaluate the effectiveness of PF-GAN staining conversion, thereby exploring the diagnostic framework’s performance in a complex tumor microenvironment.

As depicted in Figure 10, comparing the generated EBER images by PF-GAN with real EBER images demonstrates that PF-GAN achieves accurate staining conversion in complex tumor microenvironments. This is attributed to PF-GAN’s capability to discern the distinct characteristics between tumor and non-tumor cells within H&E images, and to accomplish feature alignment during the digital staining process, thereby preserving the fidelity of pathological information. It is worth noting that physical EBER staining may affect the pathologist’s intuitive analysis of non-tumor areas due to staining overflow or tissue squeezing during slide preparation, as shown in Case 1. PF-GAN can effectively avoid these problems, thereby better assisting pathologists in making accurate diagnoses. The results demonstrate the efficacy of PF-GAN in complex tumor microenvironments, enabling precise diagnostics in difficult cases and enhancing the clinical applicability of the diagnostic framework.

### 4.3. Generated Image Quality Evaluation

To evaluate the image quality of EBER images generated by different GANs, we employed several relevant image quality assessment metrics, including RMSE, PSNR, and SSIM. These metrics evaluate the generated EBER images against the real EBER images, focusing on visual perception and pixel-level differences. We utilized valis [50], a pathological image registration tool, to construct an evaluation dataset of 1434 paired images. The H&E images in this dataset serve as inputs for different GANs. The generated EBER images are then compared with the corresponding real EBER images through the abovementioned metrics. The results are presented in Table 1.

As shown in Table 1, PF-GAN achieves the lowest value on the RMSE metric (0.01174), indicating its closer pixel-level similarity to the real image, thus enhancing the quality of images generated during the image conversion process. In terms of the PSNR metric, values greater than 20 are considered acceptable (higher is better), and those exceeding 40 indicate excellent image quality. Although the images generated by the three GANs nearly reached a PSNR of 40, PF-GAN slightly leads with a score of 38.63, emphasizing its more refined handling of brightness and contrast details in images. Similarly, for the SSIM metric, all GANs scored above 0.85 (higher is better), further verifying the superiority of PF-GAN with a score of 0.8981, which is higher than the others, showcasing its excellent ability to maintain image structure and texture.

Based on these evaluation metrics, the visual quality of the EBER images produced by CycleGAN, CUT, and PF-GAN closely approximates that of real EBER images. Although there are no significant differences in quality metrics among the images generated by the three GANs, PF-GAN slightly outperforms the other two GANs across all three metrics. These results suggest that incorporating pathology–fidelity constraints can enhance the quality of image generation, ensuring that the generated images maintain high consistency with real pathological images in terms of detail and structure.

### 4.4. Consistency and Accuracy Evaluation of the PFPD Framework

#### 4.4.1. Consistency Evaluation of the Prior-Driven Image Classification System

We conducted a systematic evaluation using 17,167 EBER images selected from our training set to further validate the effectiveness of the PD-CS and its alignment with pathologist evaluations in EBER image classification. During the evaluation, we compared the classification labels generated by the system for each EBER image with the corresponding real labels provided by pathologists at the patch level, aiming to measure the consistency of the system in diagnosing EBER images.

The PD-CS demonstrated remarkable diagnostic accuracy in classifying EBER-stained images, as evidenced by the confusion matrix in Figure A1 in Appendix A. The system accurately identified 8617 out of 8686 negatives and 8480 out of 8481 positives, reflecting high precision in its classifications. These results show an overall accuracy of 99.59% and a diagnostic accuracy for positives at an impressive 99.98%. Such a performance underscores the system’s effectiveness and precision in medical diagnostics and confirms its alignment with pathologist evaluations.

#### 4.4.2. Accuracy Analysis of Generative Models within the PFPD Framework

Through the experiments described above, it has been demonstrated that the EBER images generated by CycleGAN, CUT, and PF-GAN closely resemble real EBER images, with PF-GAN producing digital EBER images of superior quality. Furthermore, the PD-CS effectively leverages the staining characteristics of EBER images and effective rules based on three progressive levels to achieve high consistency with pathologists in diagnosing EBER images. These experiments validate two critical components within the PFPD framework: digital staining and automatic diagnosis, proving the feasibility of using PFPD for diagnosing NPC. However, further experimental research is required to explore the diagnostic accuracy of different generative models when employed as the backbone of PFPD.

To better compare the performance of different generative models serving as the backbone in the PFPD framework for classifying H&E images, we employed the fully supervised classification networks ResNet50 [51] and Swin-Transformer [52] as the baseline [53]. These models were trained exclusively using the H&E images and annotated by pathologists from the training set. In Table 2, we present the performance of the fully supervised classification networks on the test dataset across various evaluation metrics, including accuracy, precision, recall, specificity, and F1-Score. As the baseline, ResNet50 demonstrated an specificity of 0.8604 in non-NPC tissue classification, while achieving a higher recall of 0.8845 in tumor classification. The overall classification accuracy was 0.8720. Moreover, Swin-Transformer performs slightly better than ResNet50, achieving a specificity of 0.8684 in non-NPC tissue classification and a recall of 0.9210 in tumor classification.

When a different GAN is selected as the backbone of PFPD, both CycleGAN and CUT achieved recalls exceeding 0.95 for H&E tumor images, surpassing ResNet50 and PF-GAN. Their performance in non-NPC tissue classification was poor, with specificities falling below 0.65. This underperformance is primarily due to the significant overstaining observed in both models, which fails to ensure pathological consistency, resulting in a high false positive rate in diagnostics. In contrast, when serving as the backbone within the PFPD framework, the PF-GAN demonstrates commendable performance in classifying non-NPC and NPC H&E images. It achieved the highest specificity of 0.8826 in non-NPC classification accuracy, significantly improving issues with false positives. Additionally, the overall classification accuracy reached a peak of 0.9040, indicating that PF-GAN ensures pathological consistency after converting H&E images to EBER images. Moreover, it attained the highest F1-Score of 0.9143, demonstrating an excellent balance between precision and recall. When PF-GAN serves as the backbone, the PFPD framework maintains a low misdiagnosis rate while effectively identifying positives, which is crucial for clinical applications. This success is attributed to the incorporation of pathology–fidelity constraints, which ensures the preservation of pathological diagnostic information during the transformation from H&E to EBER images.

In summary, when employed as the backbone within the PFPD framework, the PF-GAN utilizes automatic annotation technology to achieve a performance that surpasses the fully supervised models without the need for manual intervention. This highlights the effectiveness of PF-GAN in maintaining pathological consistency in digital staining.

### 4.5. t-SNE Visualization Analysis

By extracting the feature maps from the last layer of the GAN generator encoder and performing average pooling to reduce dimensionality, feature vectors were obtained for t-SNE analysis. Figure 11 presents the t-SNE visualizations of the CycleGAN, CUT, and PF-GAN models, used to compare the models’ performance in feature separation. Purple dots represent non-NPC data in these visualizations, and yellow dots represent tumor data. The t-SNE plots of the CycleGAN and CUT models show significant overlap between non-NPC and NPC tumor data points, indicating poor separation performance. Specifically, numerous non-NPC data points are intermixed within the tumor clusters, suggesting these two models are ineffective in distinguishing between non-NPC and tumor features.

In contrast, the t-SNE plot of the PF-GAN model demonstrates the best performance, with non-NPC and NPC tumor data points being almost wholly separated with minimal overlap. It clearly indicates that the PF-GAN model effectively captures and distinguishes between the features of non-NPC and NPC tumor data, showcasing excellent feature separation capability. The results further prove that introducing pathology–fidelity constraints can better guide the digital staining process from H&E to EBER, preserving the consistency and authenticity of pathological features.

### 4.6. Clinical Diagnostic Comparison: Pathologists vs. PFPD

To further validate the clinical efficacy of PFPD, we conducted a diagnostic comparison experiment on H&E-stained WSIs from 188 cases in the test set, comparing the results obtained from PFPD with the diagnosis provided by three pathologists. These experts include one junior and two mid-level pathologists, who independently diagnosed these H&E images without the aid of EBER and other staining images. The relevant diagnostic results and confusion matrix are detailed in Figure 12.

The results indicated that PFPD outperformed in identifying positive cases, achieving a true positive rate of 87.90%, significantly higher than all participating pathologists. Moreover, its misdiagnosis rate of 11.17% was considerably lower than that of the pathologists, demonstrating its excellent diagnostic accuracy and reliability. Specifically, the junior pathologist’s results showed a higher number of false negatives (FN-37) and false positives (FP-21), reflecting the diagnostic challenges likely due to lack of experience. In contrast, the mid-level pathologists exhibited superior diagnostic performance, with the lowest misdiagnosis rate at 17.55%. PFPD also demonstrated high precision in identifying true negatives (TN-58), exhibiting outstanding performance in the test set.

Notably, the PFPD reduces the average time required to diagnose a slide by 83.97% compared to mid-level pathologists. This significant time advantage further highlights PFPD’s potential to enhance overall diagnostic efficiency.

## 5. Conclusions and Outlook

This study proposes PFPD, a diagnostic framework for NPC, which designs a novel GAN and explainable image processing techniques to enhance diagnostic accuracy. The principal conclusions of this research are as follows:The PFPD framework, comprising PF-GAN and PD-CS, effectively utilizes the differences in image features between NPC tumor cells and non-NPC cells in H&E images to achieve digital EBER staining. Moreover, the system accurately classifies the generated images based on EBER staining specificity and existing pathological knowledge, thus facilitating the diagnosis of NPC.PF-GAN utilizes labeled EBER images—annotated by the PD-CS—and pathology–fidelity constraints during training to guide the digital staining from H&E to EBER. The experimental results demonstrate that PF-GAN guarantees higher pathological consistency in digital staining than other GANs. Furthermore, the results indicate that PFPD exceeds the diagnostic performance of fully supervised models without the need to build expensive and complex H&E-annotated data.In clinical diagnosis effectiveness analysis, the results indicate that PFPD is clinically effective, it matches the performance of mid-level pathologists, and it significantly outperforms pathologists in diagnostic efficiency. Therefore, PFPD effectively aids pathologists in efficiently and accurately diagnosing NPC.

The PFPD framework’s fully automated annotation feature eliminates the need for manual intervention, significantly improving diagnostic efficiency and reducing costs. Its versatility and extensibility facilitate its use in diagnosing various EBV-related cancers, with potential for cross-tissue diagnostics. This is expected to improve patient treatment outcomes and advance the application of deep learning in disease diagnosis.

## Figures and Tables

**Figure 1 bioengineering-11-00739-f001:**
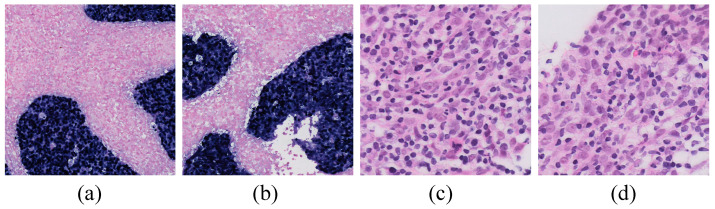
Examples of EBER and H&E images. (**a**,**b**) are examples of EBER staining, with blue representing NPC tumor cells and pink denoting non-NPC tumor cells (20× magnification). (**c**,**d**) are examples of lymphocytic infiltration into tumor cells in H&E images (80× magnification).

**Figure 2 bioengineering-11-00739-f002:**
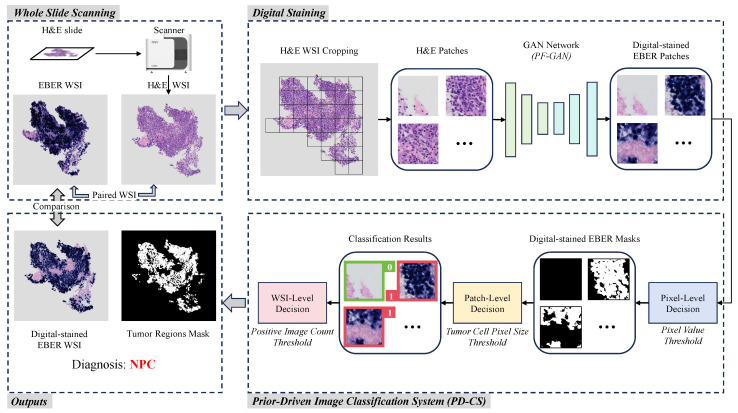
Overview of the workflow of PFPD. In PFPD, H&E patches are first cropped from the H&E WSI, and then a GAN model (PF-GAN) is used to achieve digital staining from H&E patches to EBER patches. These digital-stained patches are then input into a prior-driven image classification system (PD-CS) that applies pathological prior knowledge to guide the diagnosis of NPC (0 for EBER-negative, 1 for EBER-positive). Finally, PFPD outputs the WSI-level diagnosis result, a digital-stained EBER WSI and a mask of the tumor regions for the H&E WSI.

**Figure 3 bioengineering-11-00739-f003:**
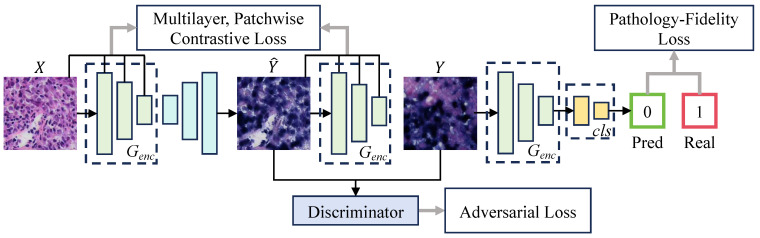
The architecture of PF-GAN. PF-GAN introduces the pathology–fidelity constraints by training the generator’s encoder to classify the target domain (0 for EBER-negative, 1 for EBER-positive), extracting global pathological features (such as tumor cell color and morphology). The constraints improve the generator’s ability to achieve accurate pathological mapping when aligning features across different domains, resulting in high-quality images with pathological consistency.

**Figure 4 bioengineering-11-00739-f004:**
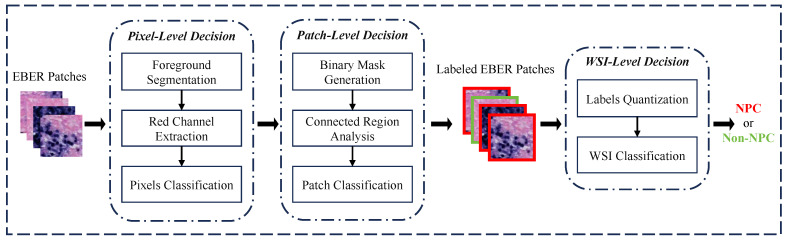
The workflow of the PD-CS. The system consists of three core decision modules: pixel-level, patch-level, and WSI-level. These modules interact via a pixel-to-WSI pathological prior knowledge to guide the diagnostic process.

**Figure 5 bioengineering-11-00739-f005:**
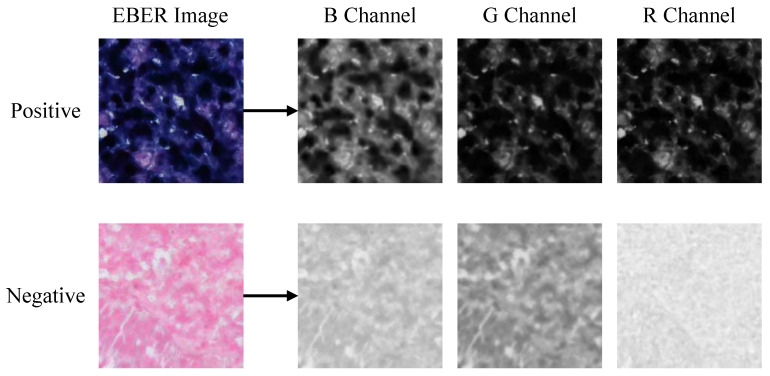
Visualization of RGB channels’ separation for positive and negative EBER images. The most significant difference between positive and negative is observed in the R channel, followed by the G and B channels.

**Figure 6 bioengineering-11-00739-f006:**
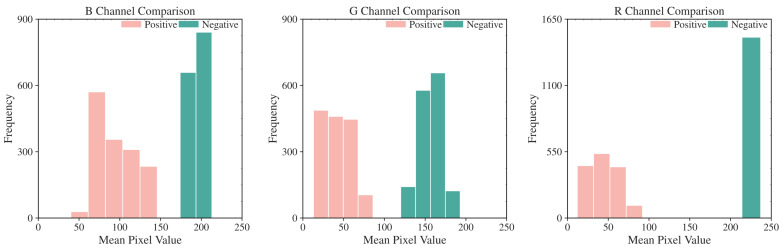
Histogram of mean pixel intensity distribution for R, G, and B Channels. R channel displays the most considerable threshold range difference between EBER-positive and EBER-negative.

**Figure 7 bioengineering-11-00739-f007:**
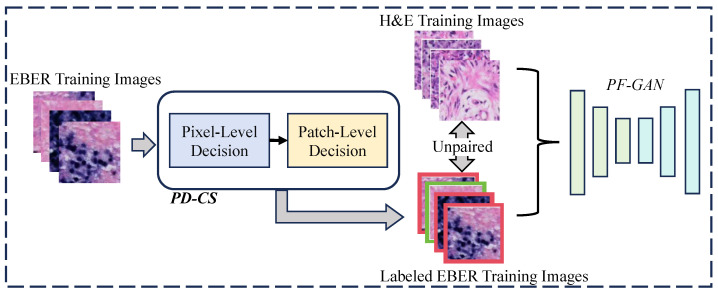
The process of automatic annotation. For PF-GAN training, EBER images are input into the PD-CS system, where annotation is accomplished using its pixel and patch modules. The labeled EBER images and unpaired H&E images are used together as training data for PF-GAN, enabling fully automatic annotation without manual intervention.

**Figure 8 bioengineering-11-00739-f008:**
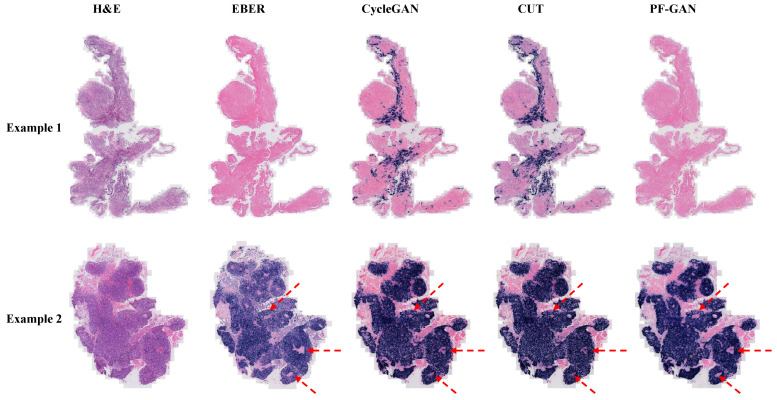
Visualization comparison of EBER staining by different GANs at the WSI level. Specifically, Example 1 is a fully negative case, Example 2 presents a case that is easily diagnosed by H&E staining. In such simple cases, PF-GAN accurately transforms large regions of tumor tissue to EBER-positive, while ensuring that small regions of non-NPC tissues remain EBER-negative (as indicated by the red arrows). This highlights PF-GAN’s remarkable capability in preserving the pathology–fidelity.

**Figure 9 bioengineering-11-00739-f009:**
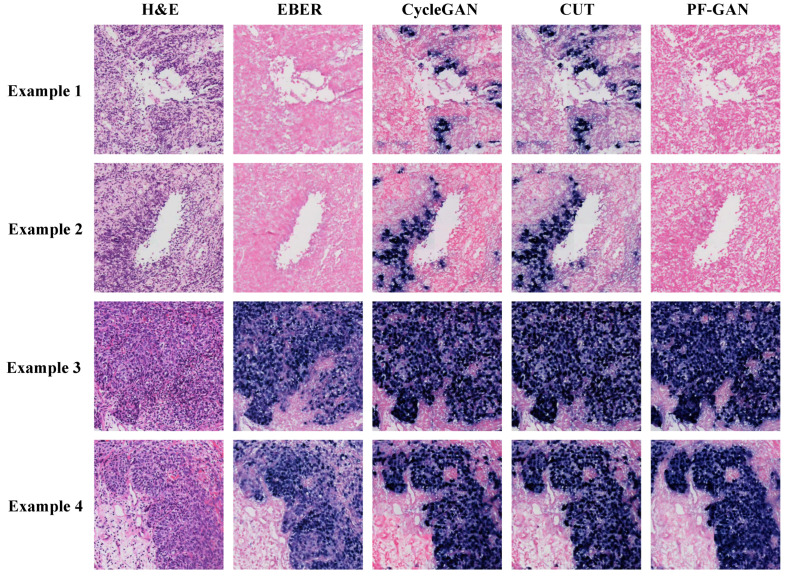
Visualization comparison of EBER staining by different GANs in organizational regions. The organizational regions shown in the figure are a 1024 × 1024 size block (20× magnification).

**Figure 10 bioengineering-11-00739-f010:**
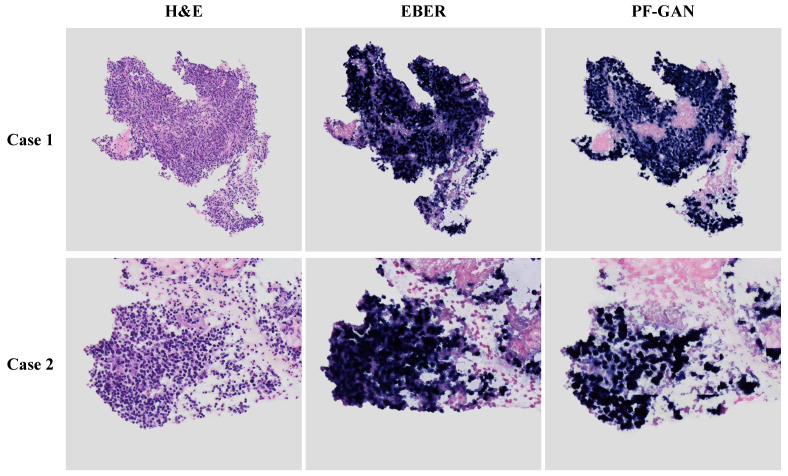
Performance of PF-GAN in difficult cases. Both Case 1 and Case 2 exhibit complex tumor microenvironments, with tumor cells infiltrated by lymphocytes.

**Figure 11 bioengineering-11-00739-f011:**
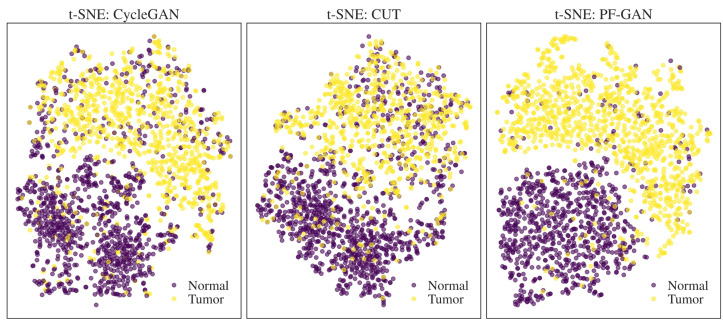
t-SNE visualization of feature separation for different GANs.

**Figure 12 bioengineering-11-00739-f012:**
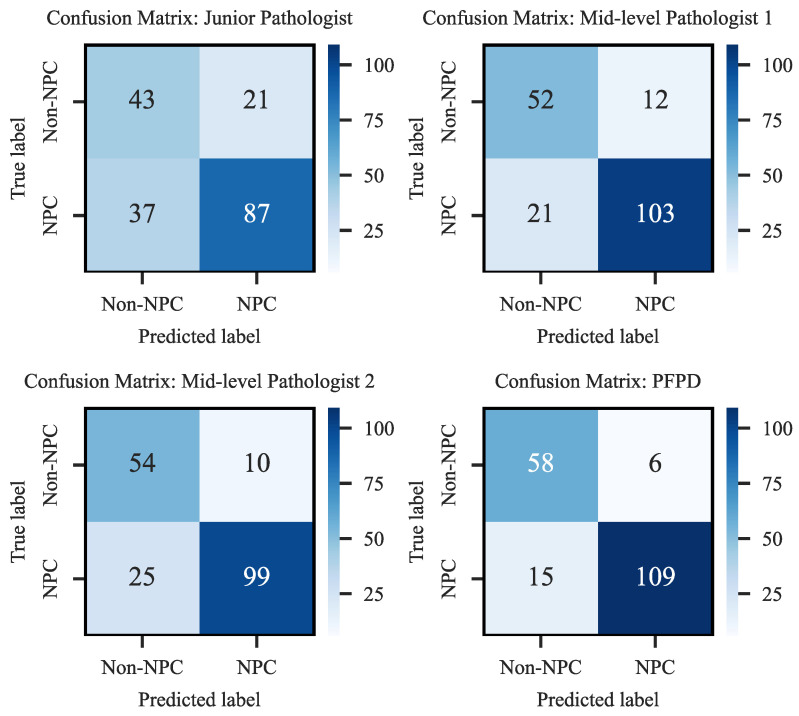
Confusion matrix of diagnostic performance.

**Table 1 bioengineering-11-00739-t001:** Comparative analysis of image quality metrics for different GANs.

Method	Image Quality Metrics		
	RMSE	SSIM	PSNR
CycleGAN	0.01228	0.8892	38.23
CUT	0.01214	0.8916	38.35
PF-GAN	0.01174	0.8981	38.63

**Table 2 bioengineering-11-00739-t002:** Comparative accuracy metrics of generative models in the PFPD Framework.

Method		Stain	Precision	Recall	Specificity	F1-Score	Accuracy
Baseline	ResNet50	H&E	0.8711	0.8845	0.8604	0.8777	0.8720
	Swin-Transformer		0.8996	0.9210	0.8684	0.9103	0.8963
Backbone	CycleGAN	EBER	0.7251	0.9530	0.6137	0.8236	0.7639
	CUT		0.7092	0.9547	0.5815	0.8139	0.7485
	PF-GAN		0.8948	0.9347	0.8826	0.9143	0.9040

## Data Availability

The datasets used during the current study are available from the corresponding author on reasonable request.

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
