# Peer review of "Automatic Annotation Diagnostic Framework for Nasopharyngeal Carcinoma via Pathology–Fidelity GAN and Prior-Driven Classification"

_bioengineering, 2024, doi:10.3390/bioengineering11070739_

Round 1
Reviewer 1 Report
Comments and Suggestions for Authors
Good research
Author Response
Response to Reviewer 1’ Comments
Dear Editor and Reviewers,
Thank you very much for providing us with the opportunity to enhance the quality of our manuscript titled “Automatic Annotation Diagnostic Framework for Nasopharyngeal Carcinoma via Pathology-Fidelity GAN and Prior-Driven Classification” (Manuscript ID: bioengineering-3107712). We are immensely grateful for the constructive and profound insights provided by the reviewers. Your feedback has been instrumental in refining our research and presentation, ensuring that our work meets the high standards expected by your esteemed journal.
In response to the comments received, we have carefully reviewed our manuscript and made detailed revisions to address each issue raised by the reviewers. These changes have been meticulously implemented to enhance the clarity, depth, and scientific rigor of our study. We have made every effort to ensure that our revised manuscript complies with the publication standards of Bioengineering.
To assist in the review process, we have highlighted all revisions in red or blue for clear visibility and quick reference. On the following pages, our point-by-point responses to the specific questions and suggestions posed by the reviewers are listed:
Comments and Suggestions for Authors:
Reviewer 1:
Comment 1: Good research.
Response: Thank you for your kind attention and care towards our work. We are grateful for your positive comment and valuable suggestions regarding the methodological aspects of our paper. In response, we have made several modifications to enhance clarity and improve the readability of the manuscript. The specific modifications are as follows:
- We have revised and added an overview of using PFPD in the NPC diagnosis at line 112 of the third page of the manuscript, highlighted in red. The changes include a detailed explanation of the workflow and fundamental principles of PFPD in diagnosing NPC. The changes are highlighted in red.
- We have added a detailed description of PF-GAN at line 145 of the fourth page of the manuscript, highlighted in red. This includes technical details, formula modifications, a comprehensive introduction to the different losses, and an emphasis on the specific role of pathological-fidelity constraints in maintaining pathological consistency in generated images. Additionally, we have explained the rationale for setting to 1.
- We have added the workflow of using PD-CS for the automatic annotation of EBER training data in PF-GAN, highlighting the automatic annotation feature of PFPD during the training process. This significantly alleviates the bottleneck of manual annotation by pathologists in the training of previous AI-based NPC diagnostic models. This modification appears on Section 3.2 (line 277, page 8), highlighted in red.
- In Section 4.3 (line 383, page 13) and Section 4.4.2 (line 426, page 14), we have added an explanation of the significance of various metrics in evaluating PF-GAN and PFPD. We have detailed how these metrics demonstrate the superiority of the framework’s performance. This modification is highlighted in red.
- We have provided a more detailed description of the dataset, clarifying that the same dataset was used to train all models to ensure fairness. Additionally, we emphasized that pathologists selected the representative cases for training to enhance data quality. This modification appears on Section 3.1 (line 260, page 8), highlighted in blue.
We appreciate the opportunity to revise our manuscript and enhance the quality of our work.
Sincerely Yours,
Corresponding author: Yonghong He
E-mail: heyh@sz.tsinghua.edu.cn

Reviewer 2 Report
Comments and Suggestions for Authors
1. Digital Staining Process and PD-CS: A detailed explanation of the digital staining process and PD-CS is needed. Specifically, there is a lack of technical details used in converting H&E patches to EBER patches.
2. Definition of PF-GAN's Loss Function (Equation 3):
>>> L = LGAN(G, D, X, Y) + LPatchNCE(G, H, X) + LPatchNCE(G, H, Y) + λPFLPF(G, cls, X, Y) Each loss component's specific role and importance in this equation need to be more clearly explained.
>>> In particular, how LPF maintains pathological consistency and why the value of λPF is set to 1 requires clarification.
3. Interpretation of Results in Table 1 and 2: The implications of the experimental results need to be clarified, and it should be made clear how these results demonstrate the framework's performance.
4. Results: "The PFPD framework’s fully automated annotation feature eliminates the need for manual intervention, significantly improving diagnostic efficiency and reducing costs."
>>> It is unclear how the fully automated annotation feature works and how much it reduces costs.
Reviewer 3 Report
Comments and Suggestions for Authors
The paper is interesting, convincing from the viewpoint of the adopted approach and as such it has a positive potential for publication. Nonetheless, at least a couple of issue remains unsolved which are:
- going for the use of ML and DL would not reduce the need of using inferential statistics and adequate tests to validate how much reliable and significative the results are. Please discuss.
- similarly the training activity in all ML/DL initiative is a very delicate process which relies on the significance, quantity and quality of the underlying datasets. This aspects is often neglectected, like in this paper. Authors should discuss it, for example with reference to paper that have already explored this kind of problems (unbalanced datasets, quality of the datasets etc ...): AA VV Is bigger always better? A controversial journey to the center of machine learning design, with uses and misuses of big data for predicting water meter failures. J Big Data, 6, 70 (2019), doi: 10.1186/s40537-019-0235-y
Round 2
Reviewer 3 Report
Comments and Suggestions for Authors
The paper is now of acceptable quality and can be published